# Spatiotemporal Variation in Saline Soil Properties in the Seasonal Frozen Area of Northeast China: A Case Study in Western Jilin Province

**Jiejie Shen [1], Yating Chen [1], Qing Wang [1,\*] and Huicheng Fu [2]**

[1]   College of Construction Engineering, Jilin University, Changchun 130026, China
[2]   Water Conservancy and Hydropower Survey and Design Institute of Jilin Province, Changchun 130021, China
\*   Correspondence: wangqing@jlu.edu.cn

**Abstract:** Due to the impact of climate change and human activities, the problem of soil salinization is increasingly prominent, posing a threat to the safety of the ecological environment and engineering construction. To understand the development tendency of soil salinization, this paper took the saline soil in Western Jilin province as the research object and carried out a long–term investigation into the basic properties of the soil at several monitoring stations. The results showed that the properties of saline soil in Western Jilin province changed regularly at the spatial and temporal scales. In the longitudinal profile, the water content, soluble salt content, and organic matter content in the soil vary greatly with the seasons at a depth range of 0–50 cm, while their changes below 50 cm are not significant. This is related to the influence depth of the external environment. Meanwhile, the content of sand is relatively stable in the depth direction, mostly between 5 and 15%, while the content of silt and clay fluctuates greatly, and there seems to be a mirror relationship between them. Along the N(W)–S(E) direction, the crystallization proportion of clay minerals gradually increases by about 28% because the relatively humid and hot climate is conducive to mineral crystallization. Over time, in the S(E) study area, the precipitation is relatively abundant, and the shallow soil is desalted due to leaching, resulting in high salt storage in the deep soil. However, in the N(W) study area, salt migrates upwards with water under the dominant effects of evaporation and freeze-thaw, leading to the accumulation of salt in shallow soil and a decrease in salt storage in deep soil. In addition, the saline soil in the study area has strong alkalinity, and the pH increases from 8.2 to 9.8 in the N(W)–S(E) direction. Overall, the soil salinization situation in Western Jilin is not optimistic.

**Keywords:** saline soil; spatiotemporal variation; water and salt migration; salt storage; salinization potential

## 1. Introduction

Saline soil is a type of soil with a high soluble salt content and is widely distributed in arid, semi–arid, and coastal areas around the world [1,2]. Particularly in China—one of the countries with severe soil salinization—the area of saline soil is approximately $3.69 \times 10^7$ hm$^2$, accounting for 4.88% of the available land area [3]. The saline soil in China has characteristics such as a wide distribution range and a large spatial span, and it covers diverse climatic conditions [4]. Concerning types of soil, it is mainly divided into sulfate saline soil in the northwest inland area, carbonate saline soil in the northeast alluvial plain, and chloride saline soil in the eastern coastal area [5]. Soil salinization not only leads to the degradation of farmland and grassland functions, but it also causes a series of engineering hazards such as building foundation corrosion, roadbed salt heave, frost heave, and collapse, seriously affecting human survival [6]. On the whole, soil salinization is a global ecological and resource problem.

The Songnen Plain is the largest distribution area of carbonate saline soil in China and is also one of the three major distribution areas of soda saline–alkali soil in the world [7].

Early tectonics and volcanic activity formed large–scale granite and volcanic belts containing a large number of alkaline minerals (such as orthoclase, plagioclase, albite, etc.), which, after long–term weathering and strong erosion, became the initial source of soda salt in the soil of Songnen Plain [8,9]. In addition, due to the low and flat terrain of the Songnen Plain, highly mineralized surface water and groundwater flow slowly, making it difficult for salt to be discharged outward [10–12]. Under the influence of climate conditions such as freeze–thaw cycles and strong evaporation, the salt in the soil continuously accumulates towards the surface with water migration, gradually forming soda saline soil mainly composed of $NaHCO_3$ and $Na_2CO_3$ [6,13,14]. Geographically, the saline soil in the Songnen Plain is mainly distributed in 13 counties in Western Jilin, among which Qian'an County, Da'an County, Zhenlai County, and Nong'an County are areas with severe soil salinization [15]. Unfortunately, from the influence of climate change and human activities, the distribution area of saline soil in Western Jilin is constantly increasing, and the degree of salinization is also deepening to varying degrees. From 1989 to 2001, the area of saline soil in Western Jilin increased from $104.31 \times 10^4$ $hm^2$ to $136.95 \times 10^4$ $hm^2$, with a significant decrease in the area of slightly saline soil, and a large increase in the area of moderately and severely saline soil [8,16]. Since the start of the 21st century, the area of saline soil continues to increase at a rate of 1–1.4% [17]. Therefore, it is imperative that we carry out an investigation and conduct research on soil salinization in Western Jilin.

In recent years, the saline soil in Western Jilin has been the focus of scholarly research. Zhou et al. [18] investigated the basic properties of saline soil in Daan County and discussed the influence of these properties on water and salt migration. Zhang et al. [19] conducted experiments on the water migration characteristics of saline soil in Western Jilin and discovered that the bound water migration in a frozen environment had a significant impact on salt migration. Shen et al. [20] studied the changes in particle size composition of saline soil in Nong'an County under freeze–thaw conditions and found that freeze–thaw cycles can cause the breakage of coarse particles and the agglomeration of fine particles. In addition, there are many studies on other characteristics of saline soil in Western Jilin, such as soil improvement [21], soil dispersion evaluation [22], and frost heave characteristics [23]. Within previous studies, it can be found that although most areas in Western Jilin have the same sedimentary background, there are significant differences in soil properties and salinization degrees across different sampling points [15–20]. This is due to regional differences in factors such as climate, terrain, hydrology (surface water and groundwater), soil management, and agricultural activities that lead to different soil evolution processes [24–26]. Nowadays, irrigation has become an increasingly common and typical measure that can improve saline soils on a large scale by leaching out salts [27,28]. However, for areas with an uneven distribution of salinization, it is difficult to uniformly wash away salts from the soil using unified irrigation. If irrigation is not reasonable, it may even cause secondary salinization disasters [29,30]. In this situation, investigation and research into the spatiotemporal variation in soil properties is not only conducive to grasping the development potential of soil salinization, but also provides theoretical support for the improvement of saline soil.

Field investigation and soil sampling are considered scientific and effective methods of information collection that can not only reflect regional geological conditions but also provide data for further research and simulation. Our team has been dedicated to the study of saline soil in Western Jilin over the past decade and has investigated soil compositions [14,18], physical and mechanical properties [31–33], soil improvement [4,21], and multi-field coupling theory [34–36]. Based on years of monitoring and experimental research in Western Jilin, the objectives of this study were to (i) summarize the characteristics of the soil properties along the longitudinal profile; (ii) investigate the differences of soil properties spatially; (iii) assess the soil salinization and alkalization status of Western Jilin; (iv) provide a reference for the improvement and management of soil salinization in Western Jilin.

## 2. Materials and Methods

### 2.1. Study Area

The study area, located in the south–central region of the Songliao Basin (Figure 1(1,2)), is a transition zone of semi–arid and semi–humid regions and is also a seasonal frozen area in Northeastern China. Spring and fall are characterized by strong winds and low humidity, with most of the evaporation (1698 mm) occurring in these two seasons. The highest temperature (35 °C) and precipitation (520 mm) are concentrated in July and August, reflecting the hot and rainy characteristics of summer. The annual average evaporation is almost treble the annual average precipitation. During winter, the lowest temperature can reach approximately −33 °C due to the influence of the Siberian cold air mass, which usually occurs between January and February. Moreover, the maximum freezing depth is close to 180 cm [15,19].

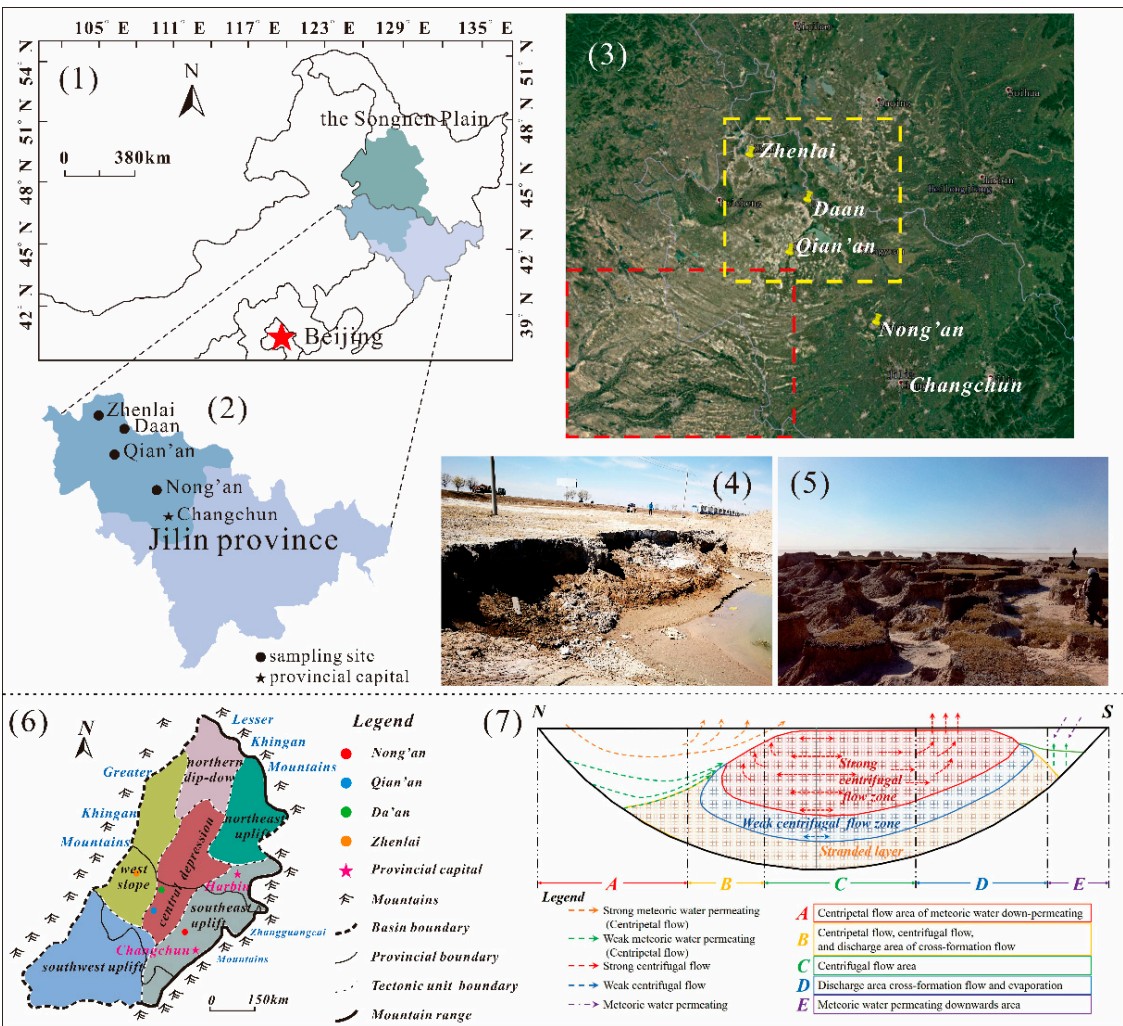

**Figure 1.** (**1**,**2**) Geographical location of the study area and monitoring stations; (**3**) google image of Songnen Plain; (**4**) slope collapse; (**5**) mud forest in Qian'an county; (**6**) the division of geological structural unit in Songnen Plain; (**7**) profile model of the hydrodynamics of Songnen Plain [37].

From the google images (Figure 1(3)), it can be seen that a large area of gray ground is distributed in the Southwest uplift and central depression of the Songliao Basin, which is the main area of soil salinization. Thereinto, $Na^+$ and $HCO_3^-$ are the main cations and anions in the salt compositions, respectively [19,32]. Due to the high content of $Na^+$ in saline soil, most of the soil is dispersive, resulting in the erosion of the slope and subgrade and the formation of mud forest landscape (Figure 1(4,5)). As shown in Figure 1(6,7), the

distribution characteristics of saline–alkali soil are closely related to the hydrogeological condition of each tectonic unit [37]. In the southwest uplift area, a group of white arc salinization lines, with an appearance reminiscent of being swept by a broom, are thought to be the salinization traces left by the decline of the groundwater level. Due to the poor drainage near the center of the basin, the surface water is mainly lost by evaporation, while the incompletely evaporated water forms some closed-flow lakes. The salinization soil in the central depression is mainly distributed around the closed–flow lakes in a porphyritic shape. Although that which is observed in Google images can reflect the salinization of surface soil to a certain extent, its appearance can also be affected by factors such as vegetation, climate, and human activities. Therefore, long–term monitoring and data collection is a scientific and accurate method for obtaining information on soil salinization.

### 2.2. Soil Samples

From 2008 to 2016, a total of 45 batches of soil samples were collected from four monitoring stations at depths ranging from 0 to 180 cm. The samples were obtained in spring (Spr., April–May, 15 batches), summer (Sum., June–August, 12 batches), and fall (Fal., September–November, 18 batches), with details listed in Table 1. Basic properties of the soils were subsequently tested, including the solid compositions (grain size composition, mineral composition, and soil organic matter content ($SOM$)) and liquid compositions (natural moisture content ($W$), soluble salt content ($S$), and pH). In addition, the non-saline soil of Changchun was taken as the control sample.

**Table 1.** The sampling station, year, season, batches, and elevations of the study soils (*: spring; *: summer; *: fall).

|  | Zhenlai | Daan | Qian'an | Nong'an |
|---|---|---|---|---|
| In 2008 | *** |  |  |  |
| In 2009 |  | *** |  |  |
| In 2012 |  |  | ** |  |
| In 2013 |  |  |  | ******** |
| In 2014 |  | *** | **** | ******* |
| In 2015 | **** | * |  | **** |
| In 2016 | ****** |  |  |  |
| Count (batches) | 13 | 7 | 6 | 19 |
| Elevation (m) | 133.5 | 129.9 | 123.6 | 193.2 |

The four saline soil sampling stations were arranged in a direction from approximately North (N) to South (S) and from West (W) to East (E), which was referred to as N(W)–S(E) in this study. The N(W)–S(E) line had significant differences based on two aspects: terrain differences in the longitude direction and climate differences in the latitude direction. The temperature increased with the decrease in latitude in the N–S direction and was also affected by the fluctuation of the terrain in the W–E direction. Therefore, the temperatures on the N(W)–S(E) line changed under the combined effect of these two factors. Figure 2a shows the temperature of five sampling sites in recent years. Taking Zhenlai and Changchun as examples, the minimum temperature during winter in Zhenlai was generally lower than that in Changchun, and the maximum temperature during non-winter months in Zhenlai was higher than that in Changchun. In other words, the monthly temperature difference in Zhenlai was larger than that in Changchun under the combined influence of longitude and latitude. Considering the other three counties, the monthly temperature difference on the line from N(W) to S(E) generally decreased. In general, the soil in the N(E) was more affected by freeze–thaw cycles than that in the S(E). Meanwhile, the monthly average precipitation in the S(E) study area was usually larger and showed an increasing trend from N(W) to S(E) (Figure 2b).

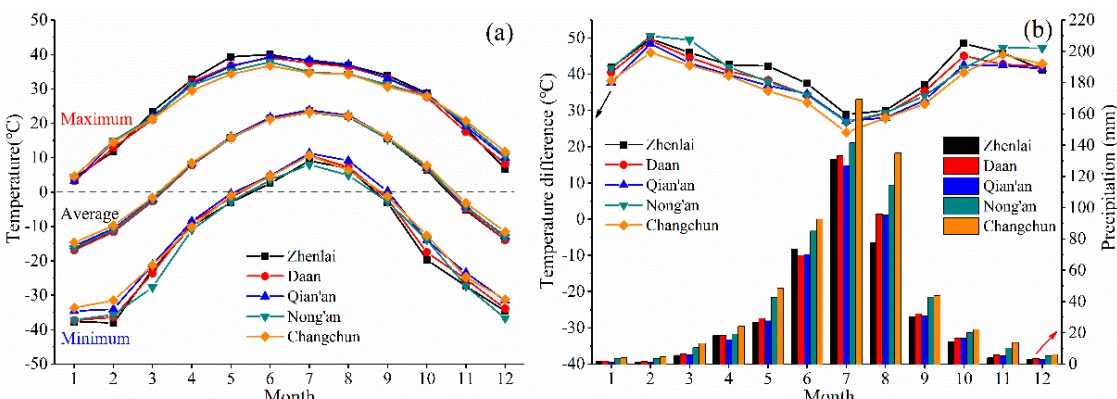

**Figure 2.** (**a**) The monthly maximum, minimum, and average temperature; (**b**) the monthly temperature difference, precipitation.

*2.3. Methods*

The experimental methods and devices are as follows. Laser particle size analysis was conducted to determine the grain size composition. The mineral composition was obtained by an X–ray diffraction (XRD) test using a D/max–2500 X–ray diffractometer. The *W* was determined by the oven drying method. The *S* was obtained by water-bath evaporation, and the soluble salt components were analyzed by titration and flame photometer methods. The potassium dichromate method was carried out to obtain *SOM*. The pH was confirmed by the colorimetric method using an electrode pH meter calibrated with standard solutions at pH 3 and 7.

## 3. Results and Discussions

*3.1. Analysis of Grain Size Composition*

Grain size composition is the most basic factor influencing the water migration and frost heave process in the soil [18]. As presented in Figure 3(1–4), fine particles, particularly silt (0.075~0.005 mm), were found to be dominant in the grain-size composition and were followed by clay (<0.005 mm), while sand (>0.075 mm) was the least. In addition, the change in sand content with depth was relatively stable, while there seemed to be a mirror relationship between the silt and the clay content. Thereinto, the seasonal change in sand content mainly occurred in shallow soil, with precipitation leaching being the main influencing factor. The content of silt and clay varied significantly with the seasons, and the depth of soil affected was deeper (Figure 3(1)).

As shown in Figure 3(2), the silt content in Nong'an County decreased gradually from 2013 to 2015, while the sand and clay content showed an overall increasing trend. On the one hand, due to relatively sufficient precipitation, fine soil particles were washed away by eluviation, resulting in a relative increase in sand content in the shallow soil. On the other hand, the freeze–thaw effect on the soil in a cold environment made some coarse particles break into fine particles at a certain depth.

In contrast, Daan was special for its intersecting climate types. Its temperature was colder, but its precipitation was relatively high (Figure 2b). In this environment, the soil was not only subject to a more intense freeze–thaw, but it was also fully affected by precipitation eluviation. Therefore, from 2009 to 2014, the sand content in Daan did not have a unified trend (it decreased in spring and increased in fall), while the content of silt and clay seemed to change in a mirror relationship (Figure 3(3)).

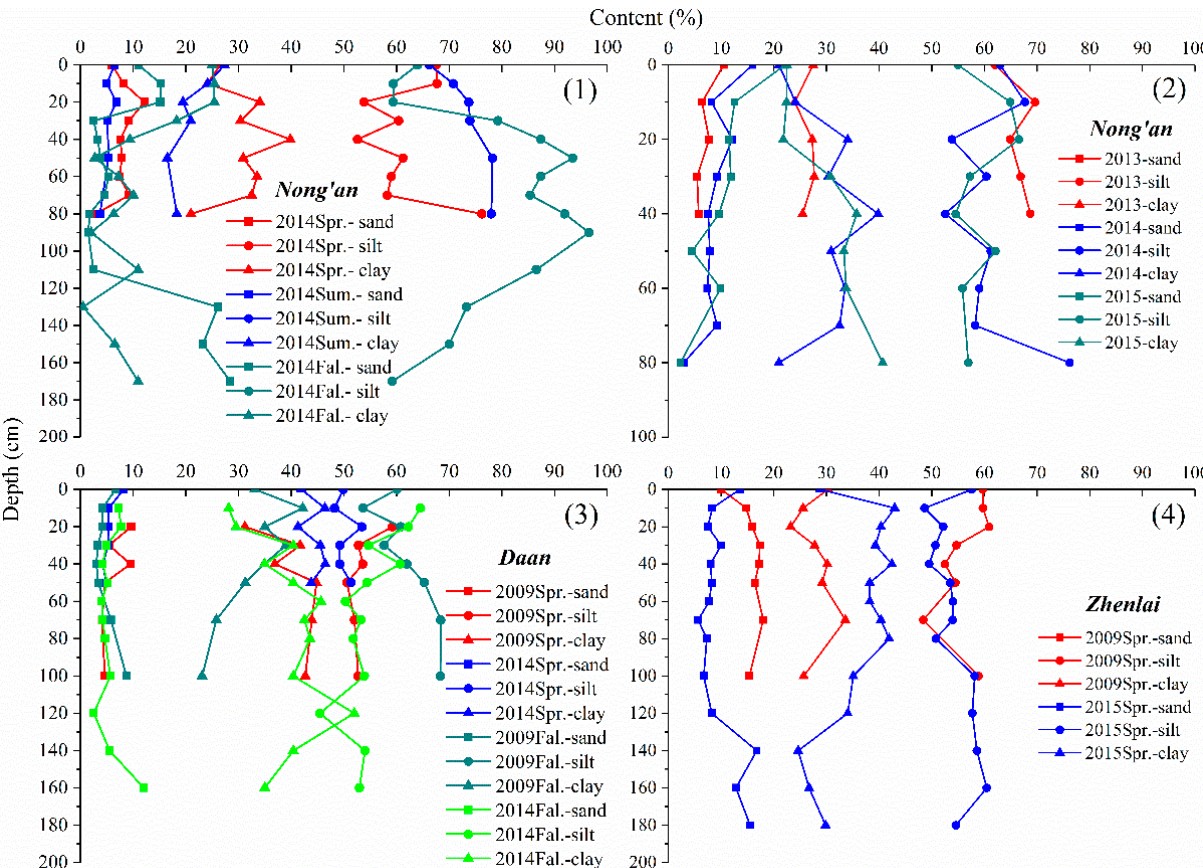

**Figure 3.** The variation in grain size composition: (**1**) Nong'an in three seasons of 2014; (**2**) Nong'an in spring from 2013–2015; (**3**) Daan in spring and fall from 2009–2014; (**4**) Zhenlai in spring from 2009–2015.

In the more arid and cold study area, the sand and silt content both decreased, and the clay content increased significantly from 2009 to 2015 in Zhenlai (Figure 3(4)). This result indicated that the freeze–thaw had a more significant impact on the crushing of coarse particles in this area.

To be more intuitive, the collected data were divided into two phases based on their timing: before 2013 (phase I) and after 2013 (phase II); the average content of the two phases in the four monitoring stations were calculated, respectively (Figure 4). Analyzing the changes in grain size across phase I and phase II, it was observed that the grain size composition of the soil was affected by temperature and precipitation at the same time. In the arid and cold N(W) area, the crushing effect of a low temperature on the coarse particles was more significant, and the clay content at a certain depth tended to increase (Figure 4(1,3)). In the semi–humid S(E) area, precipitation played a major role in changing the grain size distribution. The finer particles were washed away by the relatively abundant precipitation, resulting in an increase in sand content (Figure 4(2,4)).

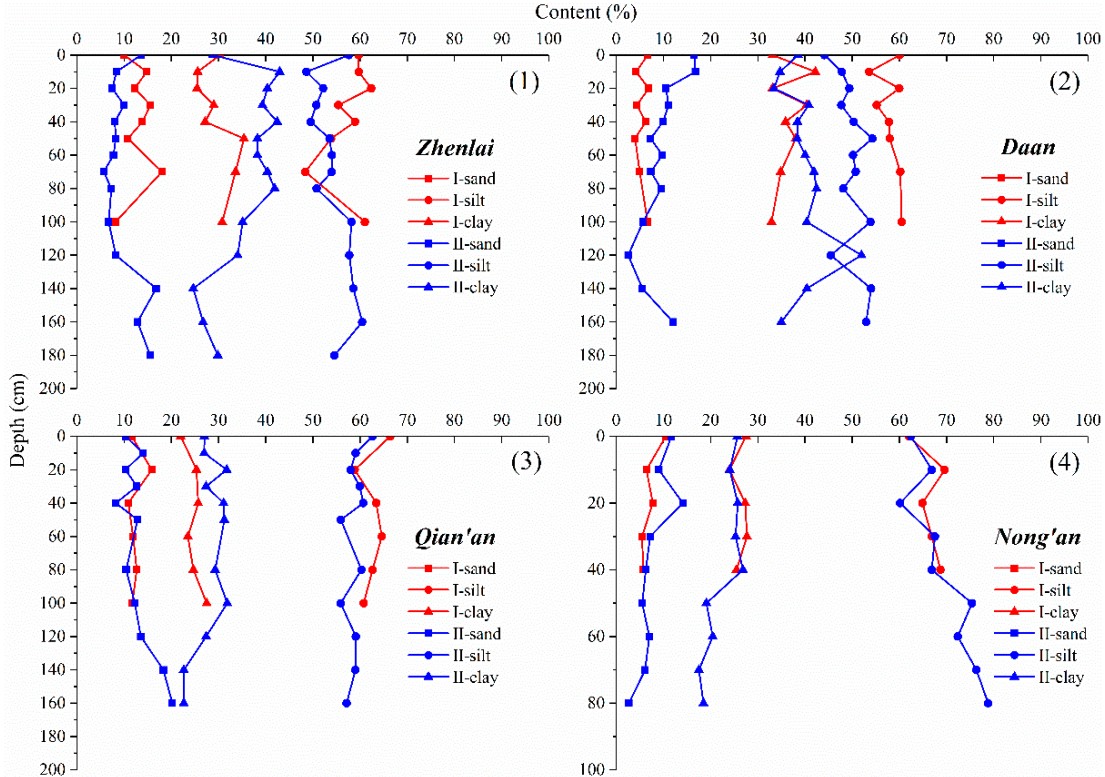

**Figure 4.** The variation in grain size composition in different phases: (**1**) Zhenlai; (**2**) Daan; (**3**) Qian'an; (**4**) Nong'an.

### 3.2. Analysis of Mineral Composition

Mineral composition, especially with secondary minerals, has a significant influence on soil properties. Figure 5 illustrates the variation trend of the mineral composition in the study area. As shown in Figure 5(1), the primary minerals accounted for more than 75% of the mineral composition, and mainly included quartz and feldspar. The secondary minerals took up a small proportion and were mainly kaolinite, illite, mix–layer I/S, a large number of microcrystals, and other clay minerals [38]. The content of the secondary minerals increased with the depth, which coincided with the variation trend of the clay particle content in the longitudinal profile. An increase in the content of clay particles or secondary minerals was beneficial for the formation of water film, but a thick water film could block the water migration channels, which was not conducive to water migration.

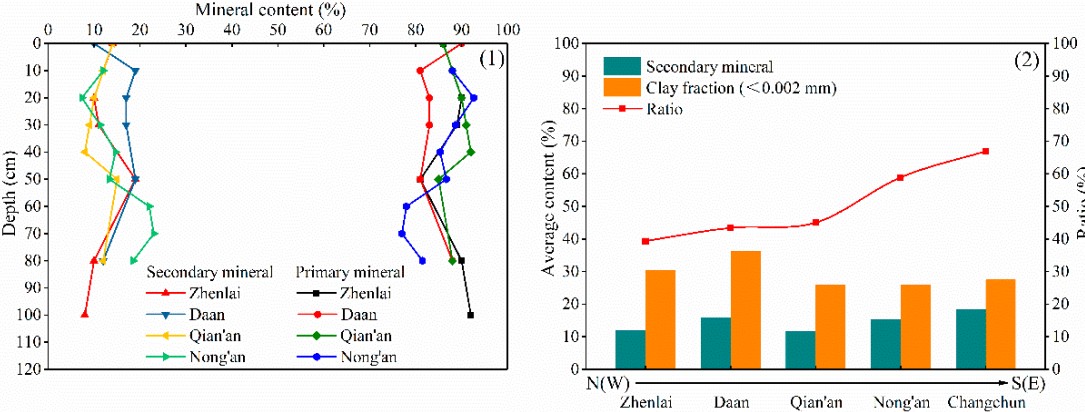

**Figure 5.** (**1**) The average content of primary and secondary minerals at different depths; (**2**) the average content of secondary minerals and clay fraction (<0.002 mm) at different monitoring stations and their ratio.

The clay fraction (<0.002 mm) in the soil was usually higher than that of the secondary minerals (Figure 5(2)), which was consistent with Chen's research findings that the structure of some minerals below the identification threshold was the reason for the content difference [15]. Therefore, the long–term effect of the arid and cold environment inhibited the crystallization of the clay minerals in this region; as a result, the crystallization of the clay minerals in the study area was poor. Meanwhile, the ratio of secondary mineral and clay fraction (<0.002 mm) showed an increasing trend in the N(W)–S(E) direction (Figure 5(2)), indicating that the relatively warm and rainy environment in the S(E) was more conducive to the crystallization of the clay minerals than that in the N(W).

### 3.3. Analysis of SOM

The *SOM* is a key reference index in soil fertility and an important carrier for plants' ability to fix $CO_2$. As shown in Figure 6(1), the *SOM* content reduced with the increase in depth within the range of 0~50 cm below the surface. After the depth exceeded 50 cm, the *SOM* content changed relatively little in the depth direction and gradually tended to be stable. Thereupon, the *SOM* content within 0~50 cm was referred to as the dynamic variation influenced by the external factors (including climate, vegetation, animal and human activities, etc.), and the *SOM* content at 50~180 cm could be regarded as the storage of each monitoring station.

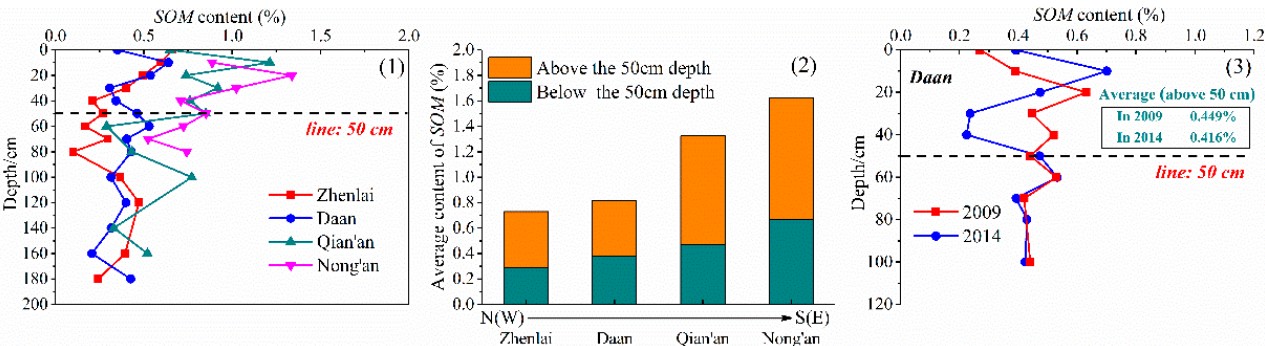

**Figure 6.** (**1**) The average *SOM* content at different depths; (**2**) the average *SOM* content at a depth range of 0~50 cm and 50~180 cm; (**3**) the average *SOM* content of Daan in 2009 and 2014.

The dynamic variation and storage of *SOM* were both larger in the S(E) than in the N(W) (Figure 6(2)): the relatively warm and rainy climate in the S(E) can not only provide a more suitable condition for vegetation and biological activities, but can also accelerate the process of chemical decomposition. Therefore, the *SOM* content reserved in the S(E) accumulated more so than that in the N(W) over the years.

Figure 6(3) compared the SOM content of Daan in 2009 and 2014. It indicates that the *SOM* content at a depth of 50~180 cm had almost no change, and the *SOM* content at 0~50 fluctuated a lot with the depth, suggesting once again that the influence of the external factors on the *SOM* content within the shallow layers was complex and polytropic. The average *SOM* content at 0~50 cm decreased by 0.033% from 2009 to 2014, indicating that the biological activities and plant growth around the sampling sites had decreased in recent years, which was not a good phenomenon for the soil quality in this area.

### 3.4. Analysis of W

Water is a solvent and carrier of soluble salt in the soil, and its content and migration mode determine the concentration and transport mechanism of salt to a great extent [39].

The *W* of different monitoring points is shown in Figure 7(1–3). In the longitudinal profile, the *W* in the soil at a depth range of 0~20 cm was low and varied greatly, then slowly increased and gradually approached stability at a depth range of 20~50 cm. When the depth exceeded 50 cm, the *W* reached an almost stable state. On the whole, the arid and windy climate had a significant impact on the *W* at a depth range of 0~50 cm, but

the turning point in the different regions was not completely the same. This was mainly related to the depth of capillary water migration influenced by evaporation. The shallow soil had a relatively lower clay content and more developed capillary pores due to leaching, resulting in a more significant upward migration of capillary water. At a greater depth, the soil contained more clay particles and the pores were gradually filled with bound water, which had strong viscosity and which hindered the upward movement of the capillary water. Therefore, the capillary phenomena became weaker in deep soil, and the change in $W$ was small [40].

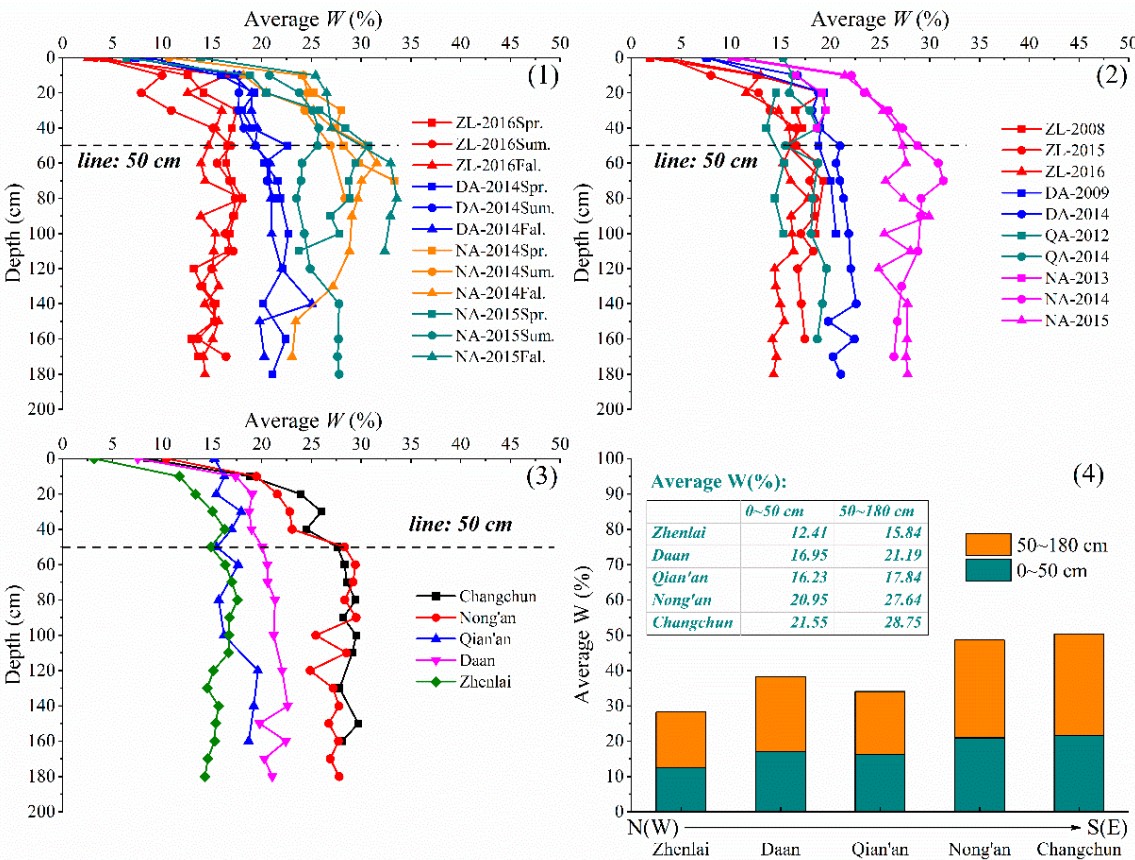

**Figure 7.** (**1**) The average $W$ in different seasons; (**2**) the average $W$ in different years; (**3**) the average $W$ of different monitoring stations; (**4**) the average $W$ within the depth range of 0~50 cm and 50~180 cm.

The variation trend of $W$ over time at different sampling points was generally consistent, but there was no absolute regularity in numerical values; it mainly depended on the weather during the days or months before sampling. Overall, the $W$ in the semi–humid area (Nong'an) was higher than that in the semi–arid area (Zhenlai), and its variation range was also greater. To provide a more detailed analysis of the changes in $W$, the average $W$ at a depth of 0~50 cm was referred to as the dynamic variation response to the local climate, and the average $W$ at a depth of 50~180 cm was regarded as the $W$ storage in this region. As depicted in Figure 7(4), the dynamic variation in $W$ tended to increase along the N(W)–S(E) line, and the storage of $W$ gradually became richer (Figure 7(4)). However, it needs to be explained that the $W$ in Qian'an was slightly lower because of the relatively higher temperature, lower precipitation, and poor water–holding capacity caused by the low content of fine particles in its grain–size composition. Hence, the $W$ was not only affected by climate factors—it was also related to the properties of the soil itself.

*3.5. Analysis of S*

It is well known that the $S$ provides an important basis for evaluating the degree of soil salinization. Therefore, the $S$ of each monitoring station in different seasons and years

was investigated, and the results are presented in Figure 8. First of all, it can be seen from Figure 8(1–4) that the *S* in the soil at different monitoring stations varied significantly with the seasons. Among them, in the arid and cold N(W) area, the sparse precipitation of the summer months cannot completely leach out the soluble salt accumulated in the shallow soil due to evaporation (Figure 8(1)), resulting in an increasing trend of the *S* in the soil with the seasons. However, in the semi–humid S(E) area, the leaching effect of precipitation gradually became more pronounced, and the *S* in the shallow soil showed fluctuating changes, decreasing in summer and increasing in spring and autumn (Figure 8(4)).

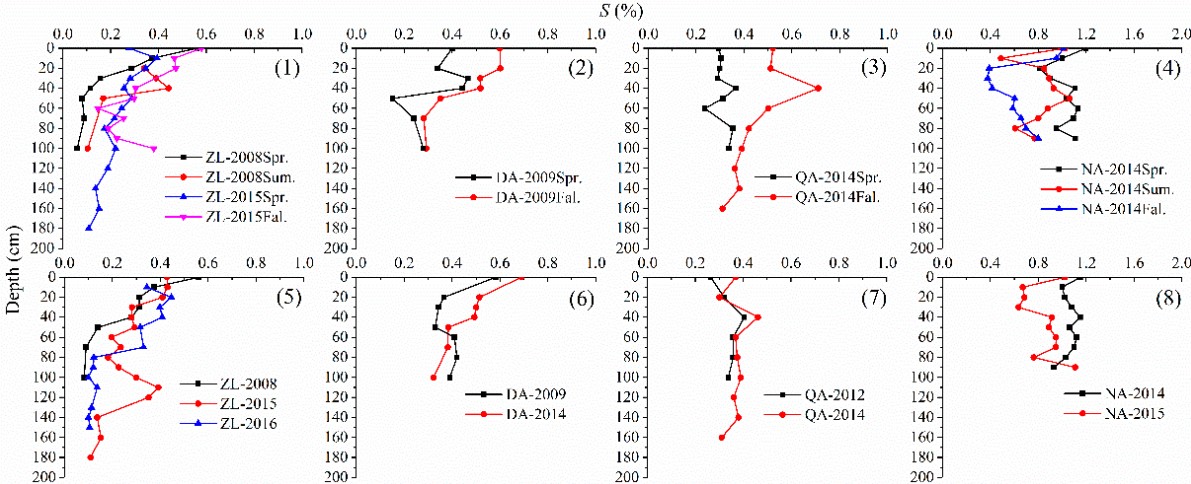

**Figure 8.** (**1–4**) The average *S* in different seasons of different monitoring stations; (**5–8**) the average *S* in different years of different monitoring stations.

Subsequently, to better understand the development tendency of soil salinization, data from each monitoring station at different seasons of the year were averaged and then compared year by year. As shown in Figure 8(5–8), the *S* in the soil at different monitoring stations demonstrated different changes with the passage of years. Therein, in the N(W) study area, under the leading role of evaporation and freeze–thaw, the soluble salt from the deep soil constantly migrated upward and accumulated in the upper soil, so the *S* in the shallow soil showed an increasing trend year by year (Figure 8(5)). Along the N(W)–S(E) direction, the leaching effect of precipitation gradually strengthened, and the accumulation of soluble salts in the shallow soil was weakened year by year (Figure 8(6,7)). In particular, the *S* in the shallow soil at the Nong'an monitoring station showed a decreasing trend year by year, indicating that the soil was in a desalting state (Figure 8(8)). From a long–term perspective, the development situation of the soil salinization in the arid and cold regions was not optimistic due to sparse precipitation and repeated freeze–thaw cycles. Moderate precipitation leaching can reduce salinity in shallow soil, which may help alleviate the degree of soil salinization.

From the averaged data of each monitoring station in Figure 9(1), it can be observed that *S* showed an overall upward trend in the N(W)–S(E) direction. In the longitudinal profile, the *S* was higher in the soil surface layer, and it decreased with an increase in depth at a depth of 0–50 cm. When the depth exceeded 50 cm, both the *S* and its variational amplitude decreased. This was related to the influence of different climates (such as evaporation, precipitation, and freeze–thaw) on *W* in the depth direction. During spring and autumn, strong evaporation caused the salt to continuously accumulate on the surface of the soil as capillary water migrated. In summer, increased precipitation led to a decrease in *S* in the shallow soil due to the downward infiltration of rainwater. In winter, under the action of temperature gradients, the salt in the deep soil migrated to the shallow soil along with the migration of the bound water. Thus, the *S* in the shallow soil varied significantly throughout the year. According to the soil water's response to climate change, the *S* was also divided into two parts (0~50 cm and 50~180 cm) in the depth direction, which were

referred to as the dynamic change (0~50 cm) and the storage (50~180 cm) of *S*, respectively. The storage of *S* at the depth range of 50~180 cm can represent the development potential of soil salinization. As shown in Figure 9(2), the dynamic variation and storage of *S* along the N(W)–S(E) direction was consistent with that of *W*, showing an increasing trend on the whole.

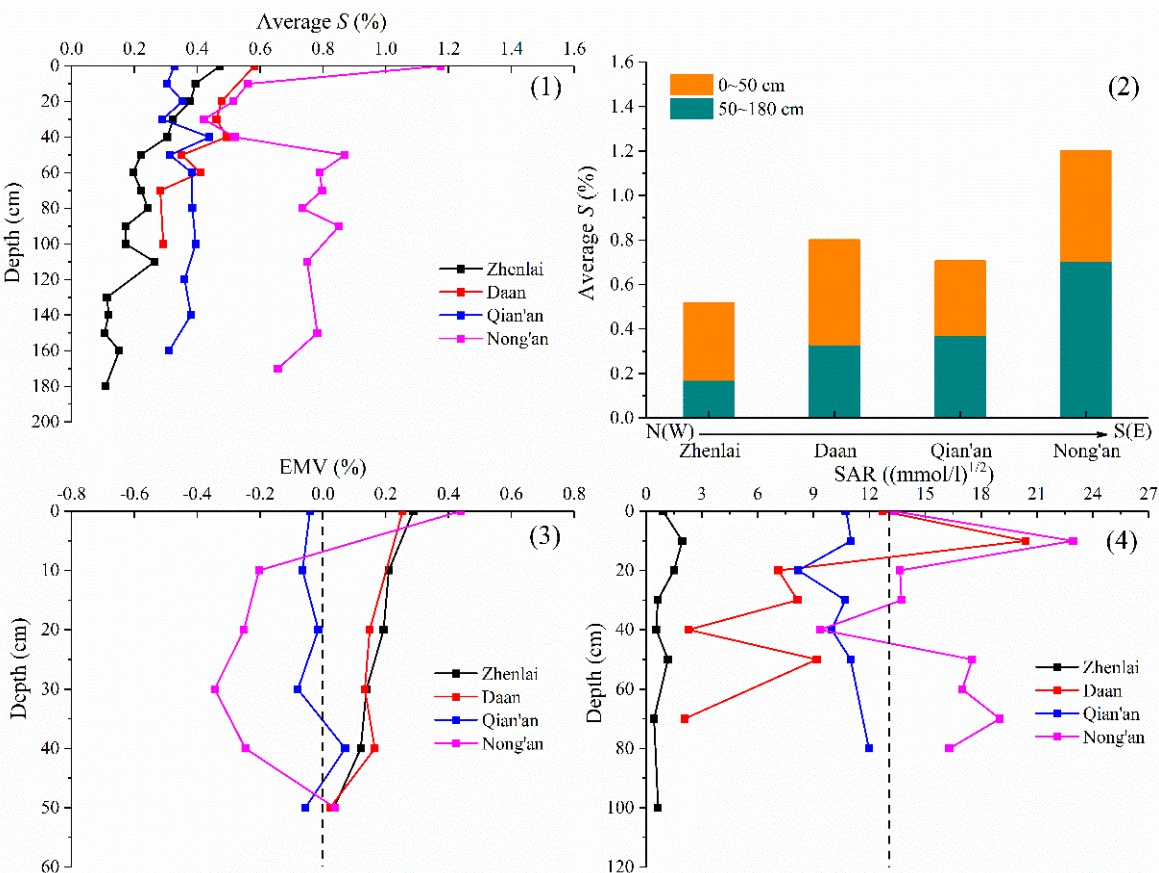

**Figure 9.** (**1**) The average *S* of different monitoring stations; (**2**) the average *S* within the depth range of 0~50 cm and 50~180 cm at different monitoring stations; (**3**) the EMV of different monitoring stations; (**4**) the SAR of different monitoring stations.

Due to the slow response of the soluble salt in the deep soil to the climate, the average *S* at 50~180 cm can be regarded as the benchmark at which to measure the migration volume at 0~50 cm quantitatively, which was referred to as the equilibrium migration volume (EMV) of the *S*. The EMV is a synthetic index for evaluating salt upward and downward salt migration, calculated as follows:

$$\text{EMV} = S_1 - S_2 \tag{1}$$

in which $S_1$ is the soluble salt content in some soil layers and $S_2$ is the benchmark level of the salt in each monitoring station. A positive EMV represented an upward migration and salt–accumulation state in the surface soil layer, and a negative EMV represented a downward migration and desalination state in the surface soil layer.

In the study area, the EMV in Zhenlai positive and gradually decreased from N(W) to S(E), with a negative EMV observed in Nong'an (Figure 9(3)). This indicated that along the N(W)–S(E) direction, the influence of evaporation and freeze–thaw on the upward migration of salt was weakened, while the influence of eluviation on the downward migration of salt was strengthened, resulting in a transformation of the soil from accumulation to desalination.

Salinization and alkalization were two aspects in the evaluation of saline–alkali soil. The total salt content was a common index of salinization, and the sodium adsorption ratio (SAR) was used to distinguish the alkalization of the soil. According to the SSSA (Soil Science Society of America), $SAR = 13 \left( \frac{mmol}{l} \right)^{\frac{1}{2}}$ was defined as the threshold value of alkaline soil and non-alkaline soil. In Figure 9(4), the SAR in the surface layer was much larger than that in the undersoil. Moreover, the SAR gradually increased in the N(W)–S(E) direction, which was similar to the trend of *S* changes in soil. It can be seen that the soil salinization process in the study area was also accompanied by alkalization, and both of them were aggravated by the increase in soluble salt content in the soil. However, from the distribution characteristics of soil salinization in the study area, the soluble salt content in most saline soil samples generally exceeded or significantly exceeded 0.3%, indicating that soil salinization was more common and serious than alkalization in Northeast China.

### 3.6. Analysis of pH

The pH of the soil can change the thickness of the diffusion layer by influencing the quantities of particle surface charge, which then affects the water migration in the soil. As shown in Figure 10(1), the pH of each monitoring station changed significantly in shallow soil, but not significantly in the deeper direction. This was related to the depth of climate influence. Furthermore, it can be found from Figure 10(2) that the average pH in the study area was greater than 8, indicating that the soil had obvious alkalization. Therein, the pH in Nong'an was the highest (about 9.8), while the pH measurements in Qian'an, Daan, and Zhenlai were close to one another and within the range of 8.2~8.5. In general, the spatial variation trend for pH was very similar to that of *S*.

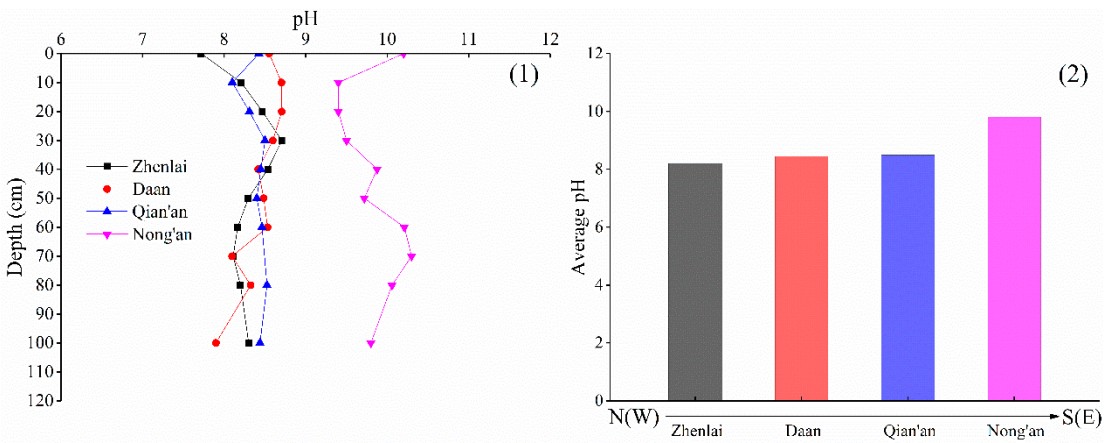

**Figure 10.** The variation in pH in the study area: (**1**) the average pH at different depths; (**2**) the average pH at different monitoring stations.

### 4. Conclusions

Based on the study of the basic soil properties and their spatiotemporal variation in Western Jilin province, the main conclusions can be drawn as follows:

(1) The soil properties *W*, *SOM*, and *S* showed obvious stratification characteristics in the longitudinal profile. (i) The first layer, ranging from 0 to 50 cm, was greatly affected by the external environment, and the *W* and *S* fluctuated significantly with the seasons. The *SOM* decreased with the decrease in humus content in relation to depth. (ii) The second layer, below 50 cm, was relatively less affected by the external environment and tended to be stable, with relatively high *W* and low *S* and *SOM*.

(2) Grain–size composition and mineral composition had no obvious stratification in the longitudinal profile. In the study area, silt was the main constituent in the grain–size composition, followed by clay, and sand was the least. The sand content changed relatively little in relation to depth, while the silt and clay content fluctuated significantly,

and there seemed to be a mirror relationship between them. Moreover, the secondary minerals accounted for less than 25% of the mineral composition and increased in relation to depth, which coincided with the variation in clay content.

(3) The soil properties in the study area changed significantly with space. In the S(E) study area, the precipitation was relatively abundant, and the average values of *W*, *SOM*, *S*, and pH in the soil were large. Under leaching, the shallow soil was desalted, and the salt storage in the deep soil was high. In the N(W) study area, evaporation and freeze–thaw played a dominant role in salt migration, and the salt state in the soil transformed from desalination into accumulation. In addition, the crystallization degree of the clay minerals tended to increase in the N(W)–S(E) direction, indicating that the relatively hot and humid climates in the S(E) were more conducive to the crystallization of the clay minerals.

(4) The process of soil salinization was also accompanied by soil alkalization in the study area, but salinization was more common and serious than alkalization. In summary, this study provided valuable insights into the spatiotemporal variation in soil properties in Western Jilin province, and the findings could help develop effective strategies for soil management in this area.

**Author Contributions:** Conceptualization, J.S. and Y.C.; data curation, J.S., Y.C. and H.F.; formal analysis, J.S. and Y.C.; investigation, J.S., Y.C., Q.W. and H.F.; methodology, J.S. and Y.C.; writing-original draft, J.S. and Y.C.; writing—review and editing, Q.W.; funding acquisition, Q.W. All authors have read and agreed to the published version of the manuscript.

**Funding:** This research was financially supported by the Key Program of International (Regional) Co-operation and Exchange of the National Natural Science Foundation of China (Grant No. 41820104001), and the Special Fund for Major Scientific Instruments of the National Natural Science Foundation of China (Grant No. 41627801).

**Informed Consent Statement:** Informed consent was obtained from all subjects involved in the study.

**Data Availability Statement:** All data are provided in this study.

**Conflicts of Interest:** The authors declare that they have no known competing financial interests or personal relationships that could have appeared to influence the work reported in this paper.

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
