# Peer review of "Spatiotemporal Variation in Saline Soil Properties in the Seasonal Frozen Area of Northeast China: A Case Study in Western Jilin Province"

_water, doi:10.3390/w15101812_

Round 1

Reviewer 1 Report

1. Abstract: put more quantitative data in this part.

2. Introduction: give the more information about distribution traits of the saline soils in China, Such as Na2CO3 in north-China, NaCl in east-China, and Na2SO4 in west-China. And then clarify the reason that why Na2CO3 is the main compostion of the salt in this region soil.

3. Materials and Methods: add the elevation of each sample site, and the effect on soil change in the following results part.  

4. Results and discussion. (1) discussion the relationship between composition of grain size and capillarity, (2)Analysis of S, whether the S means sulfur or salt? (2)discussion the relationship between water content and salt content, especially in the scope of 0-50 cm depth of the soil.

Author Response

Dear reviewers,

Thank you for your kind reviews on our manuscript entitled “Spatiotemporal variation of saline soil properties in the seasonal frozen area of Northeast China: A case study in Western Jilin Province” (Manuscript ID: Water-2302542). Special thanks to your good comments and suggestions, and we are deeply impressed by your conscientious, rigorous, high-efficiency and responsible attitude towards the evaluation work. These comments are all valuable and very helpful for revising and improving our paper.

Under the guidance in the decision letter, we have carefully and extensively revised our manuscript. Before this submission, we have consulted a native English-speaking colleague for paper revision. We hope that the corrections will meet with approval and look forward to hearing from you for any further consideration. According to your suggestions, the “Track Changes” function in Microsoft Word were used to show the modifications. The main corrections in the paper and the responses to the reviewer’s comments are attached below.

Reviewer 2 Report

The work is devoted to the study of the physicochemical composition of the soil in the riverbeds of Northeast China. The work is relevant because of the wide need for technologies and methods for calculating spatio-temporal objects. The work is built logically correctly. The mathematical and algorithmic apparatus is beyond doubt. In general, I will characterize the work as positive. However, the following changes must be made before publication.

1. The problem is not sufficiently described in the work. The introduction contains an analysis of the subject area, but there is no description of the problems and existing technical solutions. The authors propose a rather serious scientific problem, which is mainly reflected in the process of extracting artesian water. Including surface water. In this paper, the authors could say this, thereby expanding the scope of the results obtained not only to the area of rivers and surface water bodies, but also to the area of groundwater. For this reason, I recommend that the author read and refer to the work https://doi.org/10.3390/w15040770 in which the authors raise a similar topic for artesian water.

2. Also, to expand the review of the second, I recommend that you read the following works https://doi.org/10.1109/EIConRus49466.2020.9038947 and https://doi.org/10.1109/CTS48763.2019.8973323. I think that you will be able to expand and deepen your research on this topic.

3. There is no section "Discussion" in the work. The author needs to supplement the article with a section indicating the positive and negative aspects of the study. And also the author needs to explain the changes that have taken place since the day of the study. In the paper, the authors consider the range up to 2015 inclusive. Clarification is needed on what has happened since 2015 and why a wider range is not being considered.

4. Expand the list of references to 40 sources.

Conclusion. The work in question seemed to me to be a very interesting study. The authors showed fairly good scientific results, but did not narrow the scope of their application too much. I believe that these comments will allow you to improve the quality of work.

Author Response

(The authors gave the same response as above.)

Round 2

Reviewer 1 Report

The authors have revised very well accroding to my suggestion. I recommend it can be published in water journal. 

Reviewer 2 Report

Good afternoon All comments have been removed. I recommend this post.